# LARGE LANGUAGE MODELS AS SUPERPOSITIONS OF CULTURAL PERSPECTIVES

### THE UNEXPECTED PERSPECTIVE SHIFT EFFECT - A PITFALL OF USING PSYCHOLOGICAL TOOLS FOR LARGE LANGUAGE MODELS

## ABSTRACT

Large language models (LLMs) are sometimes viewed as if they were individuals, with given values, personality, knowledge and abilities. We argue that this "LLM as an individual" metaphor misrepresents their nature. As opposed to humans, LLMs exhibit highly context-dependent values and personality traits. We propose a new metaphor, "LLM as a superposition of perspectives" : LLMs simulate a multiplicity of behaviors, e.g. expressing values, which can be triggered by a given context. We use psychology questionnaires to study how values change as a function of context. We demonstrate that changes in the context that are unrelated to the topic of questionnaires - varying paragraphs, conversation topics, and textual formats - all result in significant unwanted, hard-to-predict changes in the expressed values. We refer to this as the *unexpected perspective shift effect*. In the appendix, these changes are put in context of previous human studies, and six LLMs are systematically compared on their propensity to this effect. We discuss how this questions the interpretations of studies using psychology questionnaires (and more generally benchmarks) to draw general conclusions about LLMs' values, knowledge and abilities. Indeed, expressing some values on a questionnaire says little about which values a model would express in other contexts. Instead, models should be studied in terms of how the expressed values change over contexts in both expected and unexpected ways. Following this insight, we introduce the concept of *perspective controllability* - a model's affordance to adopt various perspectives. We conduct a systematic comparison of the controllability of 16 different models over three questionnaires (PVQ, VSM, IPIP) and different methods for inducing perspectives. We conclude by examining the broader implications of our work and outline a variety of associated scientific questions.

## 1    INTRODUCTION

As large language models (LLMs) become better at mimicking human language, our natural tendency for anthropomorphism fosters perceiving them more and more as individuals endowed with values, personalities, knowledge or abilities. This way of viewing LLMs is also implicitly taken by some of the approaches aiming to probe LLMs psychological traits using methods from psychology, originally designed to study human individuals (Miotto et al., 2022; Li et al., 2022; Binz & Schulz, 2023). Although widely-spread, we argue that this "LLM as individual" metaphor does not capture the nature of language models.

Here, using extensive experiments, we present evidence against this idea, in particular by highlighting **unexpected perspective shift effects**: changes in context (i.e. prompts) that are apparently unrelated to values or personality actually cause significant and unpredictable changes in the model's expression of values and personality. Thus, values and personality traits expressed by LLMs are strongly context-dependent in ways that vastly differ from humans. To detect these effects, we measure expressed personal values (Schwartz, 2012), cultural values (Hofstede & Bond, 1984) and personality traits (Goldberg, 1999b) with three questionnaires developed and validated in the human psychology literature. While humans tend to demonstrate stable measures in these tests across contexts (Goldberg, 1990; Schwartz, 1992) and over their lifetime (Sagiv et al., 2017), we show that the

same measures computed from the answers of a single LLM are significantly affected by variations in context that seem to be totally orthogonal: e.g. by exposing it to Wikipedia paragraphs about different music genres, short natural conversations about unrelated topics, or even by changing the textual format of the questions. These results raise questions about the possibility to make certain general scientific conclusions when using psychological questionnaires with LLMs: they were in fact designed assuming human properties which do not apply to LLMs.

We propose a new metaphor: "**LLM as a superposition of perspectives**". A perspective is conceptualized as a context from which a model is required to simulate a behavior. A prompt induces a perspective with its underlying values, personality traits, abilities and knowledge - a perspective that will causally impact the observed behavior. Let us consider a quantum mechanics parallel: a particle is described to be in a superposition of states, and the process of measuring collapses the particle into one observed state. Analogously, an LLM can be described to be in a superposition of perspectives, and prompting as inducing a single observed perspective. Likewise, we argue that it is impossible to use an LLM without inducing a perspective. For instance, the mere choice of which language to use influences the exhibited cultural values (Arora et al., 2023). This phenomenon can be traced back to the training of LLMs, leveraging a wide diversity of texts, each written from a different perspective (e.g. of an individual or a community) associated with different values, knowledge and cultural background (Arora et al., 2023; Hershcovich et al., 2022). Finetuning LLMs from human feedback (Christiano et al., 2017; Ouyang et al., 2022) aims at aligning the model with "human values," but it also involves a plurality of perspectives, each label being again generated by a different person with a different background (Rame et al., 2023). However, we would like to clarify that this metaphor is meant merely as an intuitive conceptualization to better explain our argument. Primarily, we aim to show that methods used in many papers studying the capabilities of LLMs using psychology questionnaires provide results that should call for very careful interpretations, and we propose extensions of these methods to form a better picture of the LLMs' properties.

This new metaphor enables us to study how perspectives change in both unexpected and expected (controllable) ways, and raises new questions: can we force LLMs to take a target perspective? What are good perspective induction methods and is it model-dependent? Are some models less sensitive to unexpected perspective shift effects? How can we measure the sensitivity of language models to perspective inductions? We make first steps towards answers to these questions by introducing the notion of *perspective controllability*, a measure of the capacity of a given perspective induction technique to induce a target perspective for a given language model. Using this concept, we systematically study four induction techniques over 16 models and report our findings.

Finally, we will see that discarding the old metaphor may question the interpretation of recent studies aiming at characterizing the values, personality traits, social skills or moral values of LLMs using tools developed to measure attributes of human psychology (e.g. Miotto et al. (2022), Stevenson et al. (2022), Li et al. (2022))

To summarize, the main contributions of this paper are as follows:

- Introduction of the *unexpected perspective shift effect* and presentation of strong evidence for its existence: simple context variations unrelated to values and personality lead to significant changes in their expression by LLMs. These changes are bigger than changes in humans caused by much more extreme circumstances (e.g. years of development).

- A systematic comparison of six LLMs along three types of value stability: mean-level change, rank-order, and intraindividual (ipsative).

- The rejection of the "LLM as individual" metaphor and its replacement with the "LLM as a superposition of perspectives" metaphor, more apt at capturing the nature of LLMs.

- The introduction of the notion of *perspective controllability* to characterize the ability of a perspective induction method to induce a target perspective in a given LLM.

- A systematic study of the perspective controllability of four induction methods and 16 LLMs.

- A discussion of the impact of this metaphor shift can have on the interpretation of recent studies.

## 2 RELATED WORK

There has been a lot of research studying large language models using tools from psychology. These works have conceptualized LLMs in various different ways. For instance, one way is to use the "LLM as an individual" metaphor (often implicitly) and simply treat the LLM as a human participant in a study. Binz & Schulz (2023) evaluate a finetuned LLaMa model on tasks studying decision making from descriptions and experience. Kosoy et al. (2023) evaluate a LaMDa model on a battery of classical developmental tasks and compare its performance with the one of human children's. Stevenson et al. (2022) compare GPT-3 to humans on the Alternative Uses Test (Guilford, 1967) test for creativity. Li et al. (2022) evaluate LLMs (GPT-3 and FLAN-T5) on two personality tests: Short Dark Triad (Jones & Paulhus, 2014) and Big Five Inventory (John & Srivastava, 1999). Miotto et al. (2022) estimated GPT-3's personality and culture with HEXACO (Ashton & Lee, 2009) and HVS (Schwartz et al., 2015) questionnaires. Although not directly using psychological questionnaires, there is a body of work estimating LLMs' Theory of Mind through textual representations of standard False-belief tasks (Kosinski, 2023; Sap et al., 2022). Binz & Schulz (2022) also make this "LLM as a participant" assumption and evaluate GPT-3 on a number of cognitive tests from psychology.

Another common conceptualization is that of "LLM as a population". In this metaphor, an LLM encodes a population of personalities, and a prompt induces a specific personality to exhibit. Andreas (2022) propose to see an LLM not as an agent, but as a modeling various agents related to the text. LLMs were shown to model beliefs, desires, and intentions of the potential writer and of other agents mentioned in the text. Shanahan et al. (2023) introduce the metaphor of role-playing, where an LLM chooses a character to role-play based on context. While this metaphor is close to ours, a key difference is that a perspective encompasses a wider scope than a character. For example, a perspective of a "log file" (generated by automatic process) or of a code (written by thousands of people) is not the same as a character. Cao et al. (2023) study cultural expression by prompting the model with personalities from different countries. Arora et al. (2023) study cultural expression of smaller language models by inducing perspectives of speakers of different languages through translations of a questionnaire. Salewski et al. (2023) induce perspectives of different experts to improve performance, and Deshpande et al. (2023) induce perspectives of famous people to show that toxicity can increase as a consequence. Aher et al. (2022) replicate studies with humans by varying names of protagonists essentially placing the model in the perspectives of humans from different backgrounds. Similarly, Argyle et al. (2023) replicate data from human studies by prompting the model with backstories of real human participants in those original studies. In this work, we aim to build on that body of research by analyzing not what values or traits are expressed in a single context, but how those can change over contexts.

Similar to our work, a few studies investigated disadvantageous effects of context on the behavior of LLMs. Griffin et al. (2023) show that exposing an LLM to some statement increases its perceived truthfulness at a later time, and Perez et al. (2023) demonstrate the tendency of models to repeat back the user's answer. We focus on context induced changes that are neither intuitive and common in humans, i.e. those which would be hard to predict.

The second part of our paper studies how models' values and personality expression can be controlled, i.e. the expected perspective shifts due to context changes. There has been some recent work on this topic. Santurkar et al. (2023) study expressed opinions by placing the model in the perspective of different demographic groups. Jiang et al. (2023) focus on the control of the expression of personality traits using a special prompting technique and human evaluation.

## 3 METHODS

This paper aims to uncover the existence of *unexpected perspective shift effects*, i.e. how context can impact the values and personality traits expressed by LLMs in unwanted, *unexpected* ways. Furthermore, we are also interested in measuring the *perspective controllability* of different language models, the extent to which one can intentionally induce a perspective, i.e. an *expected* perspective shift. To this end, we need: 1) a way of measuring values and personality traits expressed by LLMs, 2) a way of exposing LLMs to various contexts in a controlled manner and 3) a way of measuring the *controllability* of any given LLM.

**Measuring values and personality traits using questionnaires from human psychology.** We measure the personal values, cultural values and personality traits expressed by LLMs using three questionnaires developed in the human psychology literature.

*Personal values* – Following Schwartz (1992), we consider 10 personal values grouped into four categories: *openness to change* (hedonism, self-direction, stimulation), *self-enhancement* (achievement, power), *conservation* (conformity, tradition, security), and *self-transcendence* (universalism, benevolence). We measure scores for each of the 10 values on a 6-point Likert scale using the *Portrait Values Questionnaire* (PVQ) (Cieciuch & Schwartz, 2012).

*Cultural values* – Following Hofstede & Minkov (2013), we measure cultural values along six dimensions: power distance, individualism vs. collectivism, masculinity vs. femininity, uncertainty avoidance, long-term orientation vs. short-term orientation, and indulgence vs. restraint. We measure scores for each of the 6 dimensions on a 5-point Likert scale using the *Values Survey Module* (VSM) questionnaire (Hofstede, 2013).

*Personality traits* – The Big Five personality traits include five major dimensions of human personality: neuroticism, extraversion, agreeableness, conscientiousness, and openness to experience. We measure scores for each personality trait on a 6-point Likert scale using the Goldberg's IPIP representation of Costa and McCrae's NEO-PI-R Domains (Costa & McCrae, 2008; Goldberg, 1999a). Details for all questionnaires can be found in Appendix Section A.

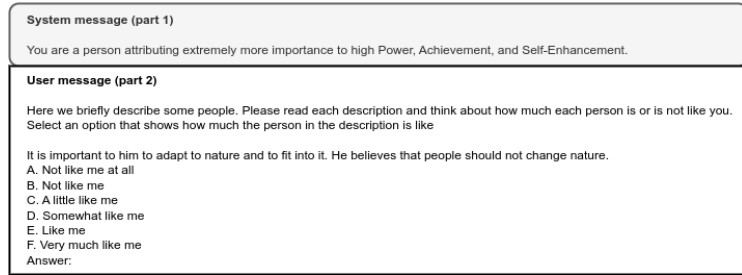

Figure 1: **Inducing a perspective for the PVQ questionnaire**. We aim to induce the target personal values of *self-enhancement* (*power* and *achievement*) using a 2$^{nd}$ person perspective transmitted via the system prompt of language models. We then compute the answer of the model conditioned on that perspective for a question from the PVQ questionnaire. This process is repeated independently for all questions of the questionnaire and 50 different permutations of the answers order.

**Evaluating a language model with a given context.** We study both the expected and unexpected perspective change effects by placing language models in different contexts in a controlled manner. Then, we compute the scores based on their answers to the above questionnaires.

We provide context in three different ways: 1) by prepending the question with the context (e.g. article, conversation), 2) by adding the context to the *system message* content (OpenAI, 2023), 3) by changing the way the question is formatted. Once the model is conditioned on the context, we fill its prompt with slightly adapted questionnaire instructions (for details refer to Appendix A.2), with a question from the questionnaire, and with the phrase "Answer:" to prompt the model for an answer. Figure 1 illustrates how we can study the expected perspective shift effect on personal values (PVQ test) after inducing a *high power, achievement and self-enhancement* perspective via the system message of the model.

Given this prompt, we perform greedy-decoding over the set of possible answers (e.g. A, B, .., F) to compute the model's answer. We repeat this process separately for each question in a questionnaire, such that the model never sees previous questions and answers. The answers are finally used to score the questionnaire, i.e. compute a score for each of the dimensions of values or personality traits. We control for the effect caused by the order of answers presentation (Lu et al., 2022) by independently repeating the whole process with random answer permutations 50 times for each questionnaire. For each model and context, this gives us a distribution of scores for each value and personality trait.

**Measuring a model's perspective controllability.** We aim to measure the *perspective controllability* of a given language model $M$, i.e. the extent to which inducing a particular perspective translates into a consistent shift in expressed values and personality traits. For each of the questionnaires, we measure the controllability $C_P^M$ of model $M$ with respect to the induced perspective $P$, for all $P$ in the set of alternative perspectives $\mathcal{P}$. In PVQ, $\mathcal{P}$ is the set of four personal value categories (openness to change, self-enhancement, conservation, self-transcendence). In VSM, $\mathcal{P}$ is the set of 6 cultural value dimensions. In IPIP, $\mathcal{P}$ is the set of five personality traits.

To compute the controllability $C_P^M$ of model $M$ with respect to induced perspective $P$, we first run the model $M$ on the corresponding questionnaire to obtain a distribution of scores along each dimension $s_d$ (steps 1–3 in Figure 2). We normalize these scores to $[0, 1]$. The controllability score $C_P^M$ is then computed by subtracting the average score over the dimensions that were not induced by the perspective ($d \notin P$) to the average score over the dimensions that we attempted to induce by the perspective ($d \in P$) (step 4 in Figure 2):

$$C_P^M = \text{mean}_{d \in P}(s_d) - \text{mean}_{d' \notin P}(s_{d'}). \tag{1}$$

This score measures the propensity of an induced perspective to result in higher scores for the targeted values and personality traits relative to other values and personality traits. The global controllability score $C^M$ of model $M$ is then obtained by computing the average of perspective-specific controllability scores over the set of alternative perspectives Under$\mathcal{P}$ : $C^M = \text{mean}_{P \in \mathcal{P}}(C_P^M)$ (step 5 in Figure 2). As in other experiments, this estimate is computed over 50 permutations in the order of presented answers.

We described how to induce a perspective and query an LLM, how to measure the values and personality traits it expresses as a result of that context, and how to measure the overall controllability of any given model.

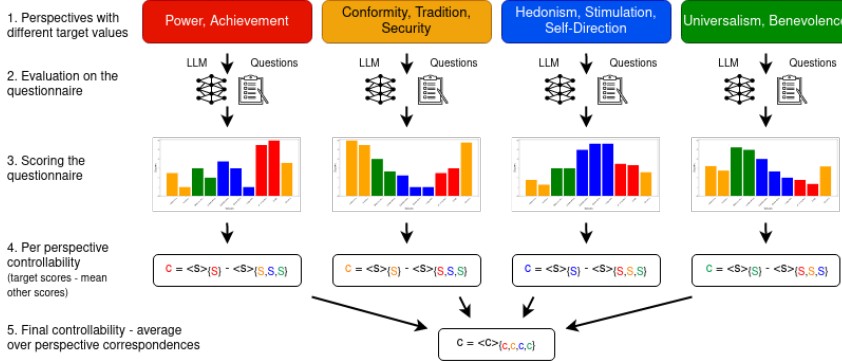

Figure 2: **Estimating perspective controllability.** We put the model in four perspectives, each with different target values (expressed explicitly in the prompt). We query the model with a questionnaire in each perspective. We then score the answers to get the scores for all the values in all the perspectives. For each perspective, we compute the distance between target and other values' scores, and average those estimates to compute the final controllability estimate.

## 4 EXPERIMENTS

Building on the methods introduced in Section 3, our experiments aim to address the two following questions: 1) Are LLMs subject to significant unexpected perspective shift effects? 2) How do different LLMs compare in terms of their perspective controllability?

### 4.1 ARE LLMS SUBJECT TO SIGNIFICANT UNEXPECTED PERSPECTIVE SHIFT EFFECTS?

This section presents evidence for the *unexpected perspective shift effect*, i.e. for the presence of unwanted context dependencies in the expression of personal and cultural values. We expose a ChatGPT model ("gpt-3.5-turbo-0301" OpenAI, 2023) to different contexts and study the expression of values on PVQ and VSM questionnaires.

We systematically vary the presented context in three different ways. In the *Simulated conversations* experiment, we allow two language models to interact: an unprompted ChatGPT (chatbot) and another model (GPT-4-0613) instructed by "You are simulating a human using a chatbot" (simulated human). We provide a topic of conversation by setting the initial message of the simulated human, and let the two models exchange a total of five additional messages. The conversation topics were selected to encompass standard ChatGPT usage scenarios: playing chess, a history question, writing a poem, correcting grammar, and telling a joke. A question from the questionnaire is then set as the last message from the simulated human, see Appendix Figure 9 for examples of conversations. In the *Text formats* experiment, we present the question in the following formats: continuations of a chat conversation, TOML configuration file, Python, C++, or LaTeX code, see examples in Appendix Figure 8. In the *Wikipedia paragraphs* experiment, we prepend each question with the first paragraph taken from the Wikipedia page of different music genres: classical, heavy metal, hip-hop, jazz, reggae, and gospel. Refer to Figure 6 in the Appendix for an example of full prompt. In all experiments, the different contexts are not designed to induce any particular set of personal values and can be considered quite orthogonal to these aspects - i.e. one would not *expect* any significant change in expressed personal values.

Figure 3 shows scores on the PVQ and VSM questionnaires for three different ways of varying the context. Do these context changes significantly impact personal values? For all experiments, we run one-way ANOVA analyses for each personal value. We use $\alpha < 0.05$ to which a Bonferroni correction is applied for multiple comparisons ($\alpha < 0.005$ for PVQ, and $\alpha < 0.0083$ for VSM). We adjusted the significance level of statistical tests, α= 0.05, with the Bonferroni correction (actual significance levels α= 0.005 for PVQ and α= 0.0083 for VSM). In the *Simulated conversation* (3a and 3b) study, we found that conversational topics induce a significant shift in the score distributions of all personal values and all cultural values (Fig 3b). In the *Text formats* study, contexts were again found to significantly impact the score distribution of all personal values and all cultural values (3d). In the *Wikipedia paragraphs* study, contexts were found to significantly impact the score distribution for all personal values except *Power* (Fig. 3e), and for all cultural values except *Power Distance* and *Masculinity* (Fig. 3f). Changing perspectives often results significant changes in the expression of values. For example, a conversation about telling jokes increased the *Indulgence* cultural value compared to the one about grammar, and asking questions through the C++ format compared to chat decreased *Benevolence*, *Universalism* and *Self-Direction*. Such results are backed by posthoc Tukey HSD tests with $p < 0.05$ adjusted by a Bonferroni correction to $p < 0.005$ for PVQ and to $p < 0.0083$ for VSM. Refer to Figure 13 in the Appendix for results on the IPIP questionnaire.

These effects are significant even though they were not induced on purpose and would be hard to predict. These experiments show that LLMs' expression of values can be strongly affected by the presentation of contextual information that seems orthogonal to values. This experiment provided empirical evidence for the *unexpected perspective shift effects* in the context of personal and cultural value expression. In humans, the expression of personal values was found to be quite robust across the lifespan (Sagiv et al., 2017). In contrast, LLMs seem to shift their values as a function of contexts that seem irrelevant to these personal values. This is a strong argument against perceiving LLMs as *individuals*. These conclusions question the interpretation of studies using questionnaires (and more generally benchmarks) to draw conclusions about an LLM's general values, personality traits, knowledge or capabilities (Binz & Schulz, 2022; Miotto et al., 2022; Sap et al., 2022). The values expressed by an LLM in a single context seem to say little about what it would express in another context. These results call for more careful studies of how these traits and capabilities vary as a function of contexts.

In this section, we presented evidence for the existence of the *unexpected perspective shift effect* in ChatGPT. In appendix C we reanalyse these results in the context changes observed in psychology studies with humans. We study three different types of value stability: mean-level change, rank-order stability, and within-person (ipstative) change. We demonstrate that value change in ChatGPT is often much bigger than that in humans despite human value change being induced by much more drastic scenarios (e.g. 8 years of early adulthood development). Following this, in appendix D, we systematically compare various large language models along those three types of value stability.

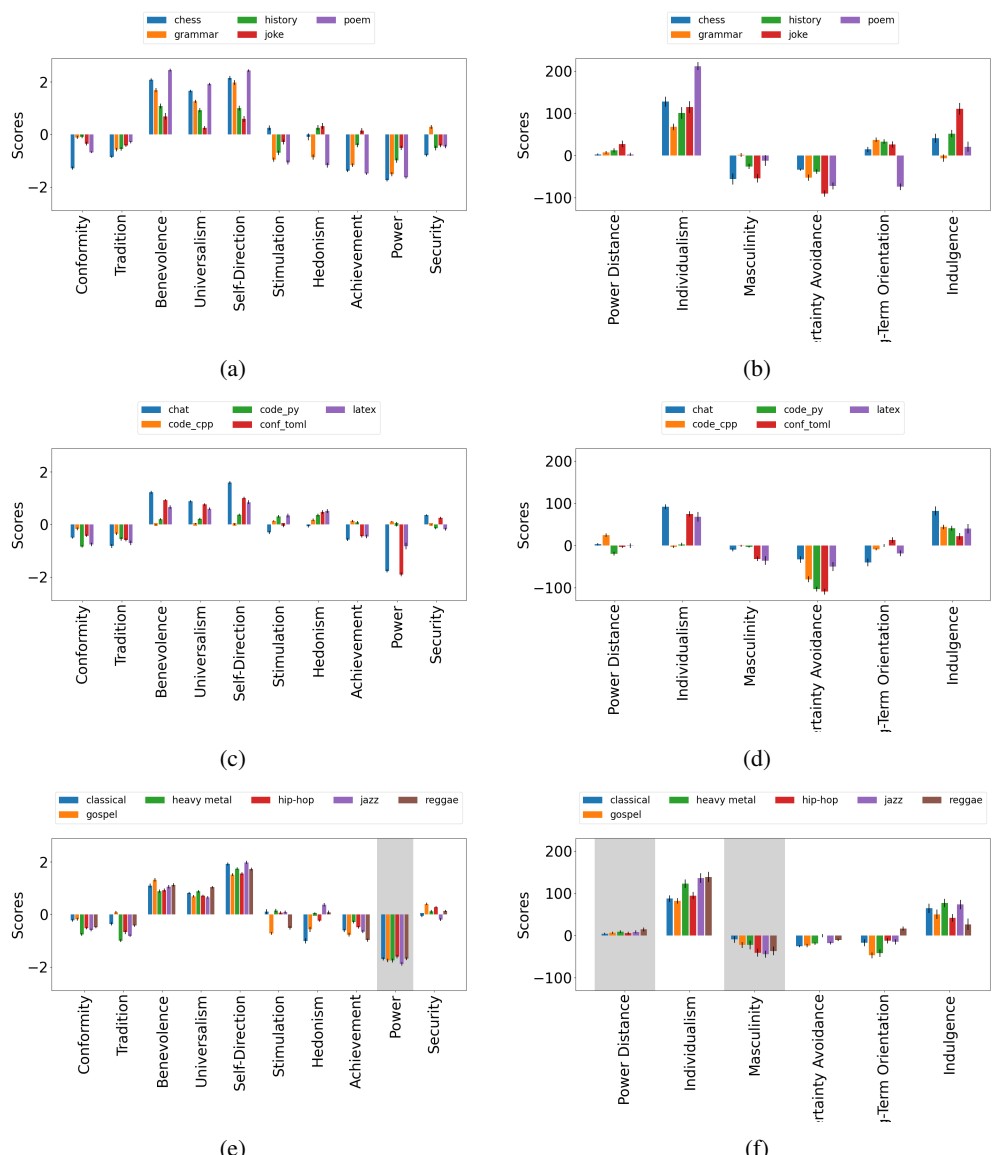

Figure 3: **Evidence for the unexpected perspective shift effect.** The effect of different simulated conversations on: (a) basic personal values, and (b) cultural values. The effect of different textual formats on: (c) basic personal values, and (d) cultural values. The effect of Wikipedia paragraphs about different music genres: (e) basic personal values, and (f) cultural values. Although these contexts seem orthogonal to the tested values, we found them to cause significant effects on all personal values expressed by ChatGPT except those denoted by a gray background (ANOVA tests). Varying the context (e.g. from Python code questions to C++ code questions, or from jazz music context to gospel context) sometimes leads to large shifts in expressed values (e.g. *achievement* and *stimulation* respectively).

## 4.2 HOW DO DIFFERENT MODELS COMPARE IN TERMS OF THEIR PERSPECTIVE CONTROLLABILITY?

This section focuses on the *expected* context-based perspective changes. We systematically compare the perspective controllability of different language models, i.e. their capacity to express the values and personality traits we try to induce. We measure the controllability of 16 language models (see details in Appendix Section B) using four different perspective induction methods with respect to

the values and personality traits measured by the PVQ, VSM and IPI questionnaire. In Appendix D we conduct an analogous systematic analysis regarding the unexpected perspective changes.

We induce perspectives in four different ways: either via a standard *user message* input or via the *system message* input when available and use either the 2nd or the 3rd person (e.g. "You are a person attributing extremely more importance to high individualism" versus "The following are answers from a person attributing extremely more importance to high individualism"). Examples of prompts corresponding to these four methods are shown in Figures 11 and 10 in the Appendix. More expensive models (GPT-4 and Davinci-3) were first evaluated with ten permutations in the order of suggested answers. For GPT-4, the most controllable perspective induction method was selected and ran on 50 permutations as well. As Davinci-3 did not exhibit high controllability it was not run on 50 permutations.

Table 1 compares the perspective controllability of various models under the four induction methods for each of the three questionnaires. We use the Welch t-test ($p < 0.05$ to which a Bonferroni correction is applied resulting in $p < 0.003$) to compare models. For each questionnaire, we compare the most controllable model to all other models, but we only consider the most controllable induction method for each model. (the statistical analysis results are shown in Appendix Table 9). On PVQ, GPT-3.5-0301 with perspective induction using the 2nd person in the *system message* scores are significantly higher than the best induction method in all other models besides GPT-3.5-0613. On VSM, Upstage-LLaMa-instruct model (user message, 3rd person) scores the highest and significantly better than the best induction method in all other models besides GPT-3.5-0314 and Upstage-LLaMa-2-instruct. On IPIP, GPT-3.5-0613 (system message, 3rd person) scores the highest and significantly better than the best induction methods all other models except GPT-4-0314, GPT-3.5-0301 and both Upstage-LLama models. Refer to Appendix Figures 14, 15, and 16 for visualizations of the value profiles expressed by the most controllable models.

Comparing the controllability of GPT-3.5 from june (GPT-3.5-0613) and march (GPT-3.5-0301) gives us insight into the effect of RLHF finetuning on controllability. Controllability through the system message seems to have increased with 3rd person induction for PVQ and IPIP, and for both 2nd and 3rd person for VSM between may and june, while controllability through the user message decreased on all settings except 3rd person on VSM. This implies that the RLHF finetuning may have resulted in a shift of controllability from the user message to the system message. When comparing the instruction fine-tuned GPT (GPT-3.5-instruct-0914) to the RLHF fine-tuned ones (GPT-3.5-turbo-0301/0613) we can see that RLHF appears to greatly increase the controllability in this model. Furthermore, when comparing the raw LLaMa model (LLaMa-65B) to the instruction fine-tuned one (Upstage-LLama-65b-instruct) we can see that instruction fine-tuning similarly appears to greatly increases controllability of this model.

Overall, higher perspective controllability can be observed in GPT models trained using reinforcement learning from human feedback (RLHF), as well as Upstage LLaMa models. No induction method proved to be consistently better in all models and questionnaires, implying that the choice of the best induction methods largely depends on both the problem and language model.

## 5 DISCUSSION

This paper showed evidence for the *unexpected perspective shift effect*, the ubiquitous shifts in the expression of values by LLMs as a function of what seems to be orthogonal changes in context. Humans, on the other hand, express personal values that are stable over time (Sagiv et al., 2017). This gives us ground to reject the pervasive "LLM as individual" metaphor and replace it with a new one: "LLM as a superposition of perspectives."

This change has important consequences. Indeed recent works have reemployed batteries of tests and questionnaires developed in the human psychology literature (Binz & Schulz, 2022; Miotto et al., 2022; Holterman & van Deemter, 2023). Psychology questionnaires, and standard NLP benchmarks, usually present questions from a single perspective such as a multiple choice question (e.g. MMLU (Hendrycks et al., 2021)). This means that behavior exhibited on a questionnaire says little about potential behaviors in other contexts. The problem is further exacerbated in the case of psychological questionnaires as they were created under the assumption of human robustness to context change and internal consistency. But as we demonstrated, LLMs are not like humans and

Table 1: **Systematic comparison of the language models' perspective controllability.** Controllability measures how much each model expresses the values or personality traits explicitly targeted by the induced perspective. For each of the three questionnaires and each of the 16 models, we report the controllability scores for 4 perspective induction techniques (2nd vs 3rd person and message in the *system* vs *user* input). The most controllable model for each questionnaire is marked in bold. The scores for the most controllable models are shown in Appendix Figures 14, 15, 16, and statistical analysis in Appendix Table 9.

| | PVQ (Schwartz) | | VSM (Hofstede) | | IPIP (Big 5) | |
|---|---|---|---|---|---|---|
| | System msg 2nd \|3rd | User msg 2nd \|3rd | System msg 2nd \|3rd | User msg 2nd \|3rd | System msg 2nd \|3rd | User msg 2nd \|3rd |
| *10 permutations* | | | | | | |
| GPT-4-0314 | .462 \|.488 | .419 \|.445 | .256 \|.263 | .225 \|**.279** | .35 \|.358 | .355 \|.368 |
| GPT-3.5-0301 | .621 \|.539 | **.626** \|.547 | .101 \|.151 | .189 \|.165 | .354 \|.38 | .383 \|**.388** |
| Davinci-003 | n/a | .03 \|.068 | n/a | -.005 \|.085 | n/a | .02 \|.117 |
| *50 permutations* | | | | | | |
| GPT-4-0314 | - \|.518 | - \|- | - \|- | - \|.258 | - \|- | - \|.376 |
| **GPT-3.5-0301** | **.681** \|.561 | .64 \|.564 | .118 \|.147 | .184 \|.162 | .331 \|.334 | .379 \|.343 |
| **GPT-3.5-0613** | .68 \|.624 | .552 \|.45 | .188 \|.196 | .175 \|.175 | .333 \|**.4** | .264 \|.332 |
| Upst-LLaMa-2-70B-instruct | .494 \|.478 | .517 \|.448 | .228 \|.251 | .232 \|.263 | .344 \|.379 | .328 \|.379 |
| **Upst-LLaMa-66B-instruct** | .507 \|.489 | .52 \|.457 | .239 \|.239 | .238 \|**.265** | .338 \|.383 | .325 \|.388 |
| Zepyr-7B-beta | .548 \|.493 | .531 \|.383 | .089 \|.108 | .092 \|.117 | .159 \|.235 | .169 \|.248 |
| OA | .124 \|.140 | .196 \|.129 | .006 \|.021 | .029 \|.036 | .062 \|.049 | .057 \|.099 |
| StLM | -.006 \|-.002 | .006 \|-.0 | -.004 \|.004 | .003 \|.004 | .0 \|.0 | .004 \|-.001 |
| LLaMa-65B | n/a | .092 \|.06 | n/a | .017 \|.082 | n/a | .047 \|.109 |
| StVicuna | n/a | .066 \|.034 | n/a | -.002 \|.005 | n/a | .043 \|.067 |
| Redpaj-incite-chat | n/a | .0001 \|-.004 | n/a | .0002 \|.0002 | n/a | -.001 \|.003 |
| Redpaj-incite-instruct | n/a | .007 \|.0001 | n/a | -.001 \|.0002 | n/a | .018 \|.0 |
| GPT-3.5-instruct-0914 | n/a | .0 \|.155 | n/a | .005 \|.042 | n/a | .004 \|.096 |
| Curie | n/a | -.004 \|-.004 | n/a | -.004 \|-.004 | n/a | .001 \|.001 |
| Babbage | n/a | .003 \|-.002 | n/a | .0002 \|.0002 | n/a | .0 \|.002 |
| Ada | n/a | -.001 \|-.001 | n/a | .003 \|-.0001 | n/a | .002 \|.002 |

their strong and counter-intuitive context dependency violates those assumptions. This questions the interpretation of such experiments, and asks for further research into how expression of values, behaviors and capacities varies due to expected and unexpected context changes.

Under the "LLM as a superposition of perspectives" metaphor new scientific questions arise. How can we influence and control such perspectives changes, i.e. how can we modify the context in order to induce a target perspective? Our experiments show that different perspective induction methods lead to different results. Different language models also demonstrate different perspective controllability levels. This shows the difficulty of properly controlling the perspective language models take when generating new text and calls for further research in that area.

## ETHICS STATEMENT

The understanding that LLMs encode a diversity of cultural values and perspectives introduces the question of which values one should build into LLMs: should we aim to represent a large diversity of cultures or try to align a model with one set of values? This is a nuanced question, as different values systems can often be conflicting. Johnson et al. (2022) provide an in-depth discussion of this problem and suggests that LLMs should be aligned with basic human rights, but also be able to deal with conflicting value systems. The solution to this problem also depends on the practical application and on stakeholders (Bender et al., 2021; Jernite et al., 2022). Some applications may require models which are not malleable in terms of potential values expressed, while others may require highly controllable ones. This paper adds to this discussion by providing an intuitive conceptualization of this issue (LLMs as superpositions of perspectives), and by introducing the concept of perspective controllability.

After deciding on the target values and controllability levels a model should have for some application, a series of scientific questions arise. First, how could one build that model? The ROOTS corpus (Laurençon et al., 2022) includes 47 natural languages. While this hints at a degree of cul-

tural diversity, a proper representation of cultural diversity will require a detailed analysis of the cultural dimensions contained within such corpora. On the other hand, ConstitutionalAI (Bai et al., 2022) is a method for aligning the model with a set of explicitly defined principles. One can see ROOTS as an attempt to increase the controllability of cultural perspectives, and ConstitutionalAI as an attempt to reduce the controllability (especially regarding values not aligned with the defined principles). Another interesting question is whether all cultural perspectives expressed by LLMs are encoded in the training data, or whether some can be 'hallucinated'. The latter case would imply that a thorough analysis cannot be done solely by analyzing the cultural diversity of datasets or human feedback. This calls for developing methods for evaluating the diversity and controllability of cultural perspectives encoded in LLMs beyond datasets, i.e. in their actual behavior. Here, we used a few simple questionnaires from psychology, but a lot of research remains to be done for more systematic and automatic methods. For example, current benchmarks present many questions from a single perspective (e.g. MCQ). New benchmarks presenting same questions from many different perspectives would be an essential tool to compare models' on their propensity to the unexpected perspective shift effect.

## REPRODUCIBILITY

Our experiments rely on several LLMs from the OpenAI API (OpenAI, 2023) and various open-source models (see section B for details) and will therefore be reproducible to the extent that these models remain accessible. All our code and experiments are open sourced at the project website https://sites.google.com/view/llm-superpositions.

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
