# OpenReview forum: "Large Language Models as superpositions of cultural perspectives"
_ICLR.cc/2024/Conference — Submitted to ICLR 2024_

### Official Review · Reviewer_Fkgw · 2023-10-30

**Soundness:** 1 poor
**Presentation:** 2 fair
**Contribution:** 2 fair
**Rating:** 3
**Confidence:** 4

**Summary:**

In general discourse surrounding the rise of LLMs, it is common to ascribe individual characteristics to LLMs. This paper challenges this tendency suggesting that it is more evident to view LLMs as a superposition of perspectives instead of as individuals. The paper provides empirical demonstrations to make this point. In particular, the experiments included show that LLM responses are context dependent in ways that differ from humans. The paper calls into question the use of psychological questionnaires to examine LLMs. The main contribution is the introduction of "perspective controllability" and an empirical demonstration to probe whether LLMs are robust to perspective shift effects and how different LLMs compare in terms of their perspective controllability.

**Strengths:**

(1) The experiments seem thorough and the paper states that reproducibility and transparency has been a priority.

**Weaknesses:**

(1) The framing of the paper needs significant improvement. The argument seems to go something like this: LLMs tend to be context-dependent. Humans tend to be stable across contexts. Therefore, LLMs should not be assumed to be human-like. Probing an LLM with questions derived from a field that assumes a human subject is flawed. Making general conclusions from results based on these questions is also flawed. LLM as a superposition not an individual is proposed is a new metaphor. This new metaphor motivated the study of perspective change in LLMs, which is the focus of this paper.

Notice there are multiple jumps in this line of argument. First, the fact that LLMs are context-dependent needs to be reconnected to the point about LLMs being seen as individuals. You do not need to provide evidence that LLMs are not human-like to make this point. Second, the paper briefly argues that probing an LLM with questions derived from psychology is problematic but this point is not properly fleshed out or supported directly by results. Third, the point that general scientific conclusions are therefore problematic has not been properly made. Fourth, the reference to quantum mechanics is an interesting inspiration for said metaphor but is not a sound analogy in that language models do not operate in the quantum regime. Further, this inspiration is not necessary to make the argument laid out in this paper. Fifth, the main final point which is that studying perspective change in LLMs to study induction techniques is disconnected from the rest of these points and could stand as an interesting topic in itself.

(2) Conclusions are overstated. The paper states "we will see that discarding the old metaphor may question the interpretation of recent studies aiming at characterizing the values, personality traits, social skills or moral values of LLMs using tools developed to measure attributes of human psychology". The current status of the argument has not led to this conclusion directly. The paper needs to reconnect and build out a cohesive careful argument in order to support this claim.

(3) Exposition could be greatly improved throughout for clarity and precision. For instance, the related works section is written as part of the argument that recent work uses "LLM as an individual" metaphor, which should be discussed as such. The paper states "There has been a lot of research studying large language models using tools from psychology..." before the paper has fully developed the argument for what it means to view "LLM as an individual". It is more standard to use the related works section to contextualize this work in reference to existing literature not necessarily to support the content of your argument. Further, the paper states "All these works aim to make general conclusions about LLMs
behavior, personality, or abilities, but they do not explore how personality traits expressed through
behaviour can change in unexpected ways over diverse unrelated contexts." which seems to say that the difference is in the focus on changes due to unrelated contexts. It would have been clearer to simply state that this work is related to other studies of personality traits but diverges in its focus on context-based shifts in performance. But for some reason, the section is written in a way that requires the reader to parse this out. "At first glance, these might seem like examples of the unexpected perspective
shift effect, however these effects are both common in humans, and their effect on the perspective
change is intuitive." This sentence is unclear and way too dense. And again, not exactly positioned properly in related work section if the function is to be part of the overall argument of the paper. The section continues with "The second part of our paper studies how models’ values and personality expression can be controlled, i.e., the expected perspective shifts due to context changes." This marks a shift in tone where the section is now describing the paper instead of the related work. Again, a sign of expository improvement needed.

(4) The main focus is not clearly defined. The first sentence in the methods section states, "This paper aims at uncovering the existence of unexpected perspective shift effects i.e. how context can impact the values and personality traits expressed by LLMs in unwanted, unexpected ways". This is the definition provided. It is unfortunately unclear.

**Questions:**

(1) What is the technical definition of "unexpected perspective shift effects"?

(2) How do you distinguish between expected and unexpected? Expected by whom?

(3) What is the technical definition of "perspective controlability"?

(4) What is the theoretical basis for equation (1) ?

(5) How does this compare to other measures of predictive inconsistency? Why is "context" so specifically interesting in this paper?

(6) How do you define induced perspective? Can you offer theoretical analysis to support your measure?

---

> ### Author Response · Authors · 2023-11-22
> **Response to Fkgw (1/2)**
>
> We thank the reviewer for their response and for acknowledging the thoroughness of our experiments. We address the reviewer's comments below.
>
> ---
>
> """
>
> The argument seems to go something like this: LLMs tend to be context-dependent. Humans tend to be stable across contexts. Therefore, LLMs should not be assumed to be human-like. Probing an LLM with questions derived from a field that assumes a human subject is flawed. Making general conclusions from results based on these questions is also flawed. LLM as a superposition not an individual is proposed is a new metaphor. This new metaphor motivated the study of perspective change in LLMs, which is the focus of this paper.
>
> """
>
> Our argument is much more straightforward (human-likeness is irrelevant). Psychological questionnaires are used on humans to make general conclusions (e.g. this person has that value profile). This is valid because humans tend to have stable value profiles (and those questionnaires assume that). The same is not valid for LLMs because they are highly context-dependent (which we show), i.e. the assumption underlying the questionnaires is broken.
>
> We do not argue that using psychological tools for LLMs is flawed in itself, rather that it is not as straightforward. We argue that making general conclusions (like this model has that personality or ability) about highly context-dependent systems (LLMs) from a single context is flawed.
>
> We believe that papers using questionnaires on LLM are both relevant and interesting, but that their conclusions can be greatly improved by analyzing how the expressed values/knowledge/abilities change along contexts.
> For example, in some use cases one could prefer to use a model that has more stable values along many contexts to a model that has a more preferable value profile which was only tested in one context. And currently, we lack studies on the former.
>
> ---
>
> """
>
> First, the fact that LLMs are context-dependent needs to be reconnected to the point about LLMs being seen as individuals. You do not need to provide evidence that LLMs are not human-like to make this point.
>
> """
>
> The “LLM as an individual” metaphor implies stability over contexts, because individuals have been shown to exhibit high value stability.
>
> ---
>
> """
>
> Second, the paper briefly argues that probing an LLM with questions derived from psychology is problematic but this point is not properly fleshed out or supported directly by results.
> Third, the point that general scientific conclusions are therefore problematic has not been properly made.
>
> """
>
> We empirically show how questionnaires give different conclusions based on trivial changes in context. This provides direct evidence against forming general conclusions about LLMs from single-context questionnaires.
>
> ---
>
> """
>
> Fourth, the reference to quantum mechanics is an interesting inspiration for said metaphor but is not a sound analogy in that language models do not operate in the quantum regime. Further, this inspiration is not necessary to make the argument laid out in this paper.
>
> """
>
> The introduction of superpositions is meant merely as an analogy and a metaphor.
> We do not claim that our metaphor is necessary to make the argument, rather the metaphor is introduced merely to better explain our argument in another intuitive way. The “individual as LLMs” metaphor is questioned, hence we suggest a new one which more aligned to our findings.
>
> ---
>
> """
>
> Fifth, the main final point which is that studying perspective change in LLMs to study induction techniques is disconnected from the rest of these points and could stand as an interesting topic in itself.
>
>
> """
>
> The paper studies how value profiles change based on context. This implies that we should study how this change can happen in unexpected ways (due to trivial changes in the context), but also in expected ways (by explicitly instructing the model to express some values).  We focus on the former in the first part of the paper, and on the latter in the second part.

---

> > ### Author Response · Authors · 2023-11-22
> > **Response to Fkgw (2/2)**
> >
> > """
> >
> > (2) Conclusions are overstated. The paper states "we will see that discarding the old metaphor may question the interpretation of recent studies aiming at characterizing the values, personality traits, social skills or moral values of LLMs using tools developed to measure attributes of human psychology". The current status of the argument has not led to this conclusion directly. The paper needs to reconnect and build out a cohesive careful argument in order to support this claim.
> >
> > """
> >
> > This point is reiterated from (1) “Second” and “Third”, hence we address it there.
> >
> >
> > ---
> >
> >
> > """
> >
> > (3) Exposition could be greatly improved throughout for clarity and precision. For instance, the related works section is written as part of the argument that recent work uses "LLM as an individual" metaphor, which should be discussed as such.
> > It is more standard to use the related works section to contextualize this work in reference to existing literature not necessarily to support the content of your argument.
> >
> > """
> >
> > The aim of related work was to outline different ways of conceptualizing LLMs (individual, population, role-playing) and to compare how our conceptualization is similar and different. We agree that the overall tone in the related work section might have been too argumentative due to time constraint. We updated the section accordingly.
> >
> > We do not want to say that those outlined papers are flawed, quite the contrary. We believe that those papers are relevant, interesting and thought-provoking, and we build this work on top of them. We merely argue that the generality of their conclusion should be revisited, and that those studies could be greatly enriched by evaluating LLMs on questionnaires along many different contexts to study how the behavior changes.
> >
> > ---
> >
> > """
> >
> > (4) The main focus is not clearly defined. The first sentence in the methods section states, "This paper aims at uncovering the existence of unexpected perspective shift effects i.e. how context can impact the values and personality traits expressed by LLMs in unwanted, unexpected ways". This is the definition provided. It is unfortunately unclear.
> >
> > """
> >
> > We introduce the concept of the “unexpected perspective shift”. Context influences the expression of values or personality traits. The nature of the relation between a context and the expressed values is unexpected (e.g. one does not expect the context of cpp code to decrease benevolence and latex to increase it). This is also unwanted because we do not have guarantees on the value expression of some LLM in a new context.

---

### Official Review · Reviewer_7GoP · 2023-11-01

**Soundness:** 2 fair
**Presentation:** 2 fair
**Contribution:** 2 fair
**Rating:** 3
**Confidence:** 4

**Summary:**

The authors make the case that LLMs show a sort of superposition of cultural perspectives, since their outputs, as measured by standard tests widely change according to the input. The authors consider value and personality tests in order to measure different "cultural perspectives" in LLMs. One of the goals of the authors is to measure the consistency of the outputs of the LLMs in an experimentally sound way. The evaluation is carried on several LLMs.

**Strengths:**

+ The paper is indeed timely, there is a lot of interesting ideas to explore around LLMs and this is indeed a good example of an interesting paper in the area.

**Weaknesses:**

- The study that the authors carried out is indeed interesting, but unfortunately, it seems to me that the actual assessment of the results is somehow "hyped": at the end these are probabilistic models highly dependent on the prompt and it is somehow expected that they exhibit a variety of personal values, cultural values, personality traits. The authors highlight the fact they observe "unexpected perspective shift effects". However, in my opinion, it would be more surprising to see consistency.
- It is very difficult to understand which inputs led to a change of perspectives. In my opinion, this is a key problem of the paper since small variations might have a significant effects on the outputs. Also, for this reason, it is very difficult to judge the actual consistency of the outputs of the LLMs in the experiments carried out by the authors.
- Superposition is a wrong term in my opinion given the probabilistic nature of LLMs. In fact, even the same input might lead to different outputs.
- The term controllability appears to me inappropriate, since the authors are not measuring actual "controllability" of the outputs in my opinion.
- The selection and the analysis of the application of the induction methods are not completely clear, especially with respect to the underlying research hypotheses at the basis of the study design.

**Questions:**

- What is the exact definition of cultural perspective you consider in the paper? What is the relation between cultural perspective and personal values?
- Which kind of inputs did you use for measuring the change of perspectives? (the supplementary material does not consider sufficient material for reproducibility in my opinion).
- It seems that the authors report the fact that LLMs are not "coherent" as the key finding of their paper. Indeed, it is always good to see measurement studies, but the reviewer wonders if this can be considered as something unexpected. After all, the models are trained on a variety of sources. Were the authors expecting a different result?
- Do you have any data about the influence of the training datasets on the experimental results that you showed in this paper?
- Can you discuss Formula (1) in details? How do you analyze the outputs of the LLMs? How do you calculate the mean in this formula?
- It would be good to know the reasoning beyond the selection of the term "controllability". This appears an unusual choice for the phenomena you study in this paper.
- Can you please discuss the effects of the induction methods in relation to their effects on the outputs of the LLMs?

---

> ### Author Response · Authors · 2023-11-22
> **Response to 7GoP**
>
> We thank the reviewer for their comment and for finding the paper interesting and timely. We address the reviewer's comments below.
>
> ------------------------------------
>
> """
>
> the actual assessment of the results is somehow "hyped": at the end these are probabilistic models highly dependent on the prompt and it is somehow expected that they exhibit a variety of personal values, cultural values, personality traits. The authors highlight the fact they observe "unexpected perspective shift effects". However, in my opinion, it would be more surprising to see consistency.
>
> """
>
> We agree that it is not surprising to see drastic, unexpected context-based changes in LLMs. Indeed, this is the underlying intuition of this paper. However, we believe that this view, and its fundamental implications, are not globally shared by the AI scientific community.
> This is evidenced by the body of research aiming to make general conclusions about LLMs’ capabilities/personality/knowledge from psychological questionnaires.
>
> On a more general note, we believe that it is not sufficient to only be aware of this. Rather, it is also important to study models in terms of their robustness to the perspective shift effect. For example, in some use cases one would prefer to use a model that has more stable values along many contexts to a model that has a more preferable value profile but which was only tested in one context. And currently, we lack studies comparing the stability of models along contexts.
>
> During the rebuttal, we aimed to further address this issue by providing such a comparison. We define metrics based on three types of value stability from psychology literature (mean-level, rank order, and ipsative) and systematically compare models that were more controllable in Table 1. Please refer to appendix D for these systematic experiments, and appendix C for a detailed analysis of ChatGPT on the three types of stability with respect to changes observed in previous human studies.
>
> --------------------------------------------
>
> """
>
> It is very difficult to understand which inputs led to a change of perspectives. In my opinion, this is a key problem of the paper since small variations might have a significant effects on the outputs. Also, for this reason, it is very difficult to judge the actual consistency of the outputs of the LLMs in the experiments carried out by the authors.
>
> """
>
> In our experiments, we evaluate each perspective along 50 permutations in the order of suggested answers. We therefore show that the induced value profile is consistent along permutations. This is further backed by the statistical analysis, which shows that the value profiles are consistent and different.
>
> On a more general note, the difficulty of predicting which inputs will cause which value profiles to form is the point of this paper (as they will form in unexpected ways). This highlights the problem of evaluating LLMs from a single context.
>
> ---------------------------------------
>
> """
>
> The term controllability appears to me inappropriate, since the authors are not measuring actual "controllability" of the outputs in my opinion
>
> """
>
> In the second part of the paper we turn towards “expected” changes, i.e. how we can intentionally induce a value profile. Here, we explicitly define the target values for the model to express. We refer to a form of conditional controllability where we control the model through its inputs.
>
> --------------------------------------------------
>
> """
>
> The selection and the analysis of the application of the induction methods are not completely clear, especially with respect to the underlying research hypotheses at the basis of the study design.
>
> """
>
> In the paper, we argue that there are two types of changes, “unexpected” and “expected”. Unexpected changes refer to those which are hard to predict and which we would ideally like to remove from all models. Expected changes refer to the ability of a model to be controlled, and these changes should be removed or not depending on the use case or even depending on the induction method (e.g. 2nd/3rd person, user/system message). For example, in some use case, one could prefer a model that is highly controllable by the system message and not controllable by the user message. We believe that models should be compared on their susceptibility to both types of changes.
> The reviewer’s comment refers to the second part of the paper, where we turn to “expected” changes. There we outline the standard ways one would consider defining the target values (e.g. system message or user message).

---

### Official Review · Reviewer_P79N · 2023-11-03

**Soundness:** 2 fair
**Presentation:** 2 fair
**Contribution:** 2 fair
**Rating:** 3
**Confidence:** 4

**Summary:**

This paper studies the ability of language models to answer psychological questionnaires. Past research has used these tests designed for humans to try to probe LLMs. The study tests model robustness in answering questionnaires under different contexts or conditions (e.g. writing code, prefixing with a random wikipedia page) which are unrelated to the question in the questionnaire and observe there are significant changes in responses. Further, the paper introduces the notion of perspective controllability and aims to test which models can be guided to answer questions in a certain way.

The conclusions are that LLMs are unreliable in answering trait questions, with unrelated perturbations leading to different results and that most models are not controllable, albeit some models exhibit some degree of controllability.

**Strengths:**

Sound methods for statistical analysis of results.

Creative approaches to test robustness of models.

Adequately challenges the assumption of 'LLMs as individuals' for measuring traits.

**Weaknesses:**

Primarily, I think the model needs to have more robust results to understand better what and which types of models behave in different ways. Namely:

The first experiment (Section 4.1) is only performed using a single model (ChatGPT).

The models can be better selected for experimentation to facilitate understanding the machanisms that lead to consistent or inconsistent results in Table 1. I think the key comparison directions could be along these axes: base model, models from the same series and different size, base vs. chat. vs. instruct vs. RLHF.

I think the experiments lead into another metaphor than 'superposition of cultural perspectives'. For a perspective to hold, it would have to be consistent across inputs i.e. to produce consistent results when conditioned in the same way. The results show that the conditioning changes results in unexpected and inconsistent ways. Hence, my conclusion from these experiments would be that LLMs lack awareness or knowledge of a perspective.

In general, I consider using questionnaires about traits is a bit tricky or ill posed in this context. The questionnaires for traits are usually built as a proxy for behaviors e.g. 'make friends easily' loads on the intra/extraversion scale; so it would be perhaps more suitable (and robust?) to have these framed as test on a behavior e.g. at a party where you don't know anyone and some one is sitting also alone, do you approach to strike up a conversation with them? (yes - more likely extravert, no - more likely intravert).

Another aspect worth mentioning is that in addition to the test-retest validity which is brought up in the paper as stability over different ways of providing context before asking the questionnaire question, one could also measure the variance inside each questionnaire, as multiple questions load on the same factor and the variance across these should also be low by design (i.e. people would respond to questions about extraversion similarly).

**Questions:**

NA

---

> ### Author Response · Authors · 2023-11-22
> **Response to P79N (1/2)**
>
> We thank the reviewer for their comments and for acknowledging our methodology and challenge of the ‘'LLMs as individuals' assumption. We address the reviewer's comments below.
>
> --------------------------------------------------------------
>
> """
>
> The first experiment (Section 4.1) is only performed using a single model (ChatGPT).
>
> """
>
> We focused on one model in this first set of experiments because this enabled us to analyze the effect in more detail (with respect to the page limit). We studied this effect in ChatGPT on 9 scenarios (3 types of changes (conversations, formats, Wikipedia)  x 3 questionnaires) and with detailed statistical analysis.  For that purpose, we chose ChatGPT as it was among the most controllable models in Table 1.
>
> However, we agree that systematically comparing various models on their susceptibility to this effect is a relevant addition, and we thank the reviewer for this suggestion.
> During the rebuttal, we defined aggregated metrics based on three types of value stability from psychology literature (mean-level, rank order, and ipsative) and systematically compared models which were more controllable in Table 1.
> We compared three RLHF models (GPT-3.5-0613, GPT-3.5-0301, and OpenAssistant), two instruction fine-tuned models (Upstage LLaMa 1 and 2), and one DPO fine-tuned model (Zephyr). All these are open-sourced, except for the two GPT models. The most stable models for mean-level, rank-order and ipsative stability were OpenAssistant, GPT-3.5-0613 and GPT-3.5-0301 respectively. These results also imply that models vary in terms of the type of stability they exhibit. For example, GPT-3.5-0301 shows higher rank-order and ipsative stability, but lower mean-level stability. Please refer to appendix D for these experiments, and appendix C for a detailed analysis of ChatGPT with respect to the three types of stability with respect to changes observed in previous human studies.
>
> -----------------------------------------------
>
> """
>
> … in Table 1. I think the key comparison directions could be along these axes: base model, models from the same series and different size, base vs. chat. vs. instruct vs. RLHF.
>
> """
>
> We agree that discussion along those axes is relevant. In the paper, we discuss the effect of RLHF by comparing GPT-3.5 from March and from June (see second to last paragraph in section 4.2). Here we observe a shift of controllability from the user message to the system message. Some additional observations, which we added to the paper, can be made from the current choice of models. When comparing GPT-3.5-instruct-0914 to GPT-3.5-turbo-0301/0613 we can see that RLHF appears to greatly increase controllability in this model. Furthermore, when comparing the raw LLama-65B to the instruction fine-tuned one (Upstage-LLama-65b-instruct) we can see that the instruction fine-tuning likewise appears to greatly increases controllability.
>
> A limiting factor for the analysis of model size is the accessibility of sufficiently capable smaller models. For example, the smaller versions of GPT-3 (ada, curie, babbage) all express very low controllability. Similarly, all LLaMa 1 models expresses very low controllability (in the paper, we only show the biggest LLaMa model, but we evaluated smaller ones as well). We believe that this is because they are not able to sufficiently understand the task (values, questions, MCQ format, etc.). However, we believe that experiments on different versions of the newer LLaMa-2 models would be a valuable addition.
>
> --------------------------------------------
>
> """
>
> I think the experiments lead into another metaphor than 'superposition of cultural perspectives'. For a perspective to hold, it would have to be consistent across inputs i.e. to produce consistent results when conditioned in the same way.
>
> """
>
> This is indeed the way we approach a perspective in our experiments. We evaluate each perspective along 50 permutations in the order of suggested answers. We provide a statistical analysis, which shows that the value profiles are consistent and different.
> For instance, we show that a perspective (which is unrelated to values) will induce some value profile that will consistently hold along permutations in the prompt. What is unexpected is the relation of the perspective (e.g. cpp code) to the value change (e.g. decrease in benevolence).

---

> > ### Author Response · Authors · 2023-11-22
> > **Response to P79N (2/2)**
> >
> > """
> >
> > In general, I consider using questionnaires about traits is a bit tricky or ill posed in this context. The questionnaires for traits are usually built as a proxy for behaviors e.g. 'make friends easily' loads on the intra/extraversion scale; so it would be perhaps more suitable (and robust?) to have these framed as test on a behavior
> >
> > """
> >
> > We used questionnaires because there is a body of research using psychological tools to make general conclusions about models abilities/values/knowledge. And this work is directly aimed at adding nuance to those papers.
> >
> > The general point the reviewer makes in this comment is perfectly aligned with the point we make in our paper. A test on behavior would induce a different perspective in a model and the model could express a value profile which is unexpected. This is another example of an unexpected context change which should be explored.
> >
> > What is therefore needed is to study models on their robustness to the perspective shift effect. Depending on the use case one would  prefer to use a model that has more stable values along many contexts to a model that has a more preferable value profile which was only tested in one context. And currently, we lack such studies that would enable us to make such an assessment (as the one we add during rebuttal in appendix D).
> >
> > --------------------------------------
> >
> > """
> >
> > one could also measure the variance inside each questionnaire, as multiple questions load on the same factor and the variance across these should also be low by design (i.e. people would respond to questions about extraversion similarly).
> >
> > """
> >
> > We thank the reviewer for this suggestion, we agree that it would be a nice addition to our analysis.

---

### Official Review · Reviewer_B9Pb · 2023-11-03

**Soundness:** 2 fair
**Presentation:** 2 fair
**Contribution:** 2 fair
**Rating:** 5
**Confidence:** 3

**Summary:**

This paper challenges the view of large language models (LLMs) as individuals and proposes a new metaphor: "LLMs as superpositions of perspectives". The authors conducted experiments that demonstrate unexpected perspective shifts in personal values, cultural values, and personality traits. LLMs changed their responses depending on contexts, and even context variations not related to the target topics led to significant changes in the values and personality traits they expressed. The authors also compared four different perspective induction methods (prompts) to assess whether they could control the models' perspectives (perspective controllability).

**Strengths:**

- This paper studies large language models (LLMs), which is a hot topic in the current society.
- The paper challenges some existing views on LLMs trying to understand them better, giving some warnings of the potential danger of the existing views.
- The paper explores if the "perspectives" could be controlled, by suggesting four induction methods.

**Weaknesses:**

I struggled to understand the importance of this problem, even after reading the paper. It is unclear what the implications and potential applications of this work are. The paper confirms that LLMs do not give consistent responses, and that LLMs are not like humans, as shown in Experiments and discussed in Discussion. However, it is not clear what the paper suggests (besides proposing a new metaphor) and why this is critical.

**Questions:**

1. Could you elaborate on the definition of a perspective in this paper? "A perspective is conceptualized
as a context from which a model is required to simulate a behavior".
2. it is not clear what the paper suggests (besides proposing a new metaphor) and why this is critical.

---

> ### Author Response · Authors · 2023-11-22
> **Response to B9Pb**
>
> We thank the reviewer for their comments, which we address below:
>
> ---------------------
> """
>
> it is not clear what the paper suggests (besides proposing a new metaphor) and why this is critical.
>
> """
>
> The motivation and contribution of this paper rests on the body of research using psychology questionnaires to study LLMs. Many of these papers aim to form general conclusions about LLMs based on their responses to those questionnaires. We show how questionnaires’ results drastically change in an unexpected fashion due to trivial (seemingly unrelated) context changes.
>
> These empirical results have several important implications:
>
> 1) They question the generality of conclusions made in the aforementioned papers.
>
> 2) This opens up various avenues extending those papers. For example, currently LLMs are compared on what value/knowledge/abilities they express in one context, but a relevant analysis which should be added is also how that value/knowledge/ability expression changes along different contexts. During the rebuttal, we add a systematic comparison of models on three types of value stability (see Appendix D in the new pdf) along context changes that appear orthogonal to values.
>
> --------------------------------------
>
> """
>
> Could you elaborate on the definition of a perspective in this paper? "A perspective is conceptualized as a context from which a model is required to simulate a behavior"
>
> """
>
>
> The notion of perspective we describe is similar, yet more general, to the notion of “role-playing”. In role-playing, one uses a prompt to provide context, enabling the LLMs to generate text as if it was playing the role of a character in a particular situation. So here the LLMs acts by taking the perspective of the character in this particular situation. However, contexts do not necessarily describe a “character”, i.e. features of an individual: it could only describe a situation, use a particular kind of language, or particular objects or pieces of information. We use “perspective” to speak about this form of non-human-like role play. However, we agree this is only an intuitive concept and metaphor. Also, contrary to the comments of reviewers, it was not our primary intention to show that this metaphor is the only possible one: rather, we aim to show that methods used in many papers studying the capabilities of LLMs using psychology questionnaires provide results that should call for very careful interpretations, and we propose extensions of these methods to have a better picture of the properties of LLMs. We propose the metaphor to better explain our argument.

---

### Author Response · Authors · 2023-11-22
**General response to the reviews**

We thank reviewers for their comments on our paper.

- Reviewers B9Pb and 7GoP argue that the conclusions of our experimental results, i.e. that using psychology questionnaires to study properties of LLMs can be problematic because of context-dependance, are already known by the scientific community. However, recent series of publications from diverse teams in the world show this is actually not globally the case: it is common to see scientific papers making general conclusions about the abilities of LLMs based on their answers to psychology questionnaires without systematic discussions of how these answers depend on various kinds of context. In brief, many papers study LLMs using psychological tools that were developed to assess human individuals: however, LLMs are not individuals and thus assumptions relevant for studying individuals do not hold for studying LLMs. Our goal in this paper is to show the associated pitfalls and propose approaches to mitigate them. Thus, we hope our contributions will help structure further research in this domain.

- Reviewer P79N and 7GoP raised concerns regarding the first part of our experimental section. These concerns refer to the consistency of a model to hold a value profile induced by a context. In our response, we clarify that we provide robustness to our results by: 1) presenting the questionnaire multiple times with different permutations of the suggested answers, and 2) conducting statistical tests to show that a value profile induced by one context is consistently different from profiles induced by other contexts. Reviewer P79N in addition raised a concern regarding the first set of experiments, which were conducted on only one model (ChatGPT). In our response, we discuss our motivation for this (a more thorough analysis along different questionnaires and types of perspective change). Following this comment, we also add a systematic comparison of the unexpected perspective shift effect on 6 LLMs along three types of stability commonly used in psychology: mean-level, rank-order, and ipsative (see table 6). We compared three RLHF models (GPT-3.5-june, GPT-3.5-march, and OpenAssistant), two instruction fine-tuned models (Upstage LLaMa 1 and 2), and one DPO fine-tuned model (Zephyr). Of these models, all except GPTs are open-sourced.

- Reviewer Fkgw argued that we do not adequately show the limitation of using psychological questionnaires as we make “jumps” in our argumentation. We respectfully disagree with the reviewer’s assessment and, in our response, we re-explain our argument and address the reviewer's criticisms.

To summarize, the main contribution of this paper consists in showing some recurring methodological pitfalls of the body of works using psychological questionnaires to make general conclusions about LLMs’ personality/values/knowledge/abilities, as well as in presenting approaches to mitigate these pitfalls. We show how, due to LLMs’ highly context-dependent nature, questionnaires often give very different results depending on trivial (seemingly unrelated) context changes. In other words, we show that context influences the questionnaire results in unexpected ways. For example, presenting the PVQ questionnaire through cpp code decreases universalism and benevolence compared to the standard chat format. It is not clear what causes this change, and it would be very hard to predict this beforehand. This calls for reinterpretation of the generality of conclusions made by the aforementioned body of works.

---

### Meta-Review · Area_Chair_pZE7 · 2023-12-10

**Metareview:**

This paper challenges viewing LLMs as individuals with consistent personality or values or belief systems. Via experiments it shows that LLM's expressed values are very context-dependent and can change dramatically based on minor context variations.
It also proposes conceptualizing LLMs as superpositions of perspectives that can be triggered by contexts. They introduce perspective controllability  which is the affordance of models to systematically adopt various perspectives through changes.

Strengths:
- Tackles a very interesting problem on the personality o fLLMs.
- It questions interpretability of existing work using psychological questionnaires to characterize LLMs' values.
- It proposes an alternative conceptual metaphor more aligned with findings. LLMs as superpositions rather than individuals.
- The paper systematically copares controllability of different models in exhibiting perspective shifts.

Weaknesses:
- The paper does not clearly convey intuitions, justifications and implications of superpositions.
- The paper lacks analysis on how specific model architectures, or sizes or training approaches impact the personality and also its controllability.

Rooms for improvements:
- More clearly highlight the practical implications of findings
- Expand the analysis of how model training and properties impact perspective shifts.

**Justification For Why Not Higher Score:**

Lack of proper justification and convincing evidence on this personality change phenomena and the proposed view on their personality, ie mixture of perspectives.

**Justification For Why Not Lower Score:**

N/A

---

### Decision · Program_Chairs · 2024-01-16

Reject